# Thermal and Colorimetric Parameter Evaluation of Thermally Aged Materials: A Study of Diglycidyl Ether of Bisphenol A/Triethylenetetramine System and Fique Fabric-Reinforced Epoxy Composites

**DOI:** 10.3390/polym15183761

**Published:** 2023-09-14

**Authors:** Michelle Souza Oliveira, Artur Camposo Pereira, Henry Alonso Colorado, Natalin Michele Meliande, Juliana dos Santos Carneiro da Cunha, André Ben-Hur da Silva Figueiredo, Sergio Neves Monteiro

**Affiliations:** 1Department of Materials Science, Military Institute of Engineering-IME, Praça General Tibúrcio, 80, Urca, Rio de Janeiro 22290-270, Brazil; camposo@ime.eb.br (A.C.P.); nmeliande@gmail.com (N.M.M.); julianacunha@ime.eb.br (J.d.S.C.d.C.); abenhur@ime.eb.br (A.B.-H.d.S.F.); snevesmonteiro@gmail.com (S.N.M.); 2CComposites Laboratory, Universidad de Antioquia UdeA, Street 70, n° 52-21, Medellin 050010, Colombia; henry.colorado@udea.edu.co; 3Modeling, Metrology, Simulation and Additive Manufacture Section, Brazilian Army Technology Center-CTEx, Avenida das Américas, 28.705, Guaratiba, Rio de Janeiro 23020-470, Brazil; 4Department of Polymeric Materials, Federal University of Amazonas, Avenue General Rodrigo Octavio Jordão Ramos, 1200, Manaus 69067-005, Brazil

**Keywords:** natural fiber, DGEBA/TETA system, thermally aged, fique-fiber woven fabric, thermogravimetric derivative, differential thermal analysis, thermomechanical analysis, CIE Lab

## Abstract

The main modifications of thermal and colorimetric parameters after thermal aging of DGEBA/TETA system (plain epoxy) and fique-fiber woven fabric-reinforced epoxy composites are described. As a preliminary study, thermal analysis was carried out on epoxy matrix composites reinforced with 15, 30, 40 and 50% fique-fiber woven fabric. After this previous analysis, the 40% composite was chosen to be thermally aged, at 170 °C. Three exposure times were considered, namely, 0, 72, 120 and 240 h. Samples were studied by thermogravimetric analysis (TGA), differential thermal analysis (DTA), differential scanning calorimetry (DSC), thermomechanical analysis (TMA) and colorimetry analysis. Significant color changes were observed after thermal aging combined with oxidation. It was also found that the thermal behavior of the plain epoxy showed greater resistance after thermal exposure. By contrast, the composites were more sensitive to temperature variations as a result of thermal stresses induced between fique fibers and the epoxy matrix.

## 1. Introduction

Nowadays, there is a worldwide trend in search of resources that will benefit society, not only in the creation of new products, but also in the opportunity to create work and generate income through the development of new technologies with sustainability. Brazil does not disregard this principle; therefore, the current interest in composites with natural fibers, whether of vegetable, mineral or animal origin, is mainly due to its growing concern with the preservation of the environment and socialization [1]. Vegetable fibers, usually defined as natural fibers, are actually widely used as reinforcements in polymeric composites due to their low cost and biodegradability, which is consistent with the current appeal for environmental preservation and the use of renewable materials. Due to their inherent hydroxyl-rich nature, plant fibers are particularly useful in thermosetting systems. The main applications of these composites are in civil construction, furniture industries, packaging and the automotive sector. For its application, a fiber-reinforced composite (FRC) must have certain properties capable of withstanding physical, mechanical and chemical degradation processes caused by agents such as mechanical action, heat, oxygen atmosphere, and light, among others [2,3,4,5,6,7,8].

With this in mind, thermoxidative and mechanochemical transformations in a natural fiber and its polymer matrix composites are inevitable after use and, consequently, reduce the material’s performance [9]. These transformations involve partial destruction of the composite structures, making it necessary to find ways to minimize these processes. Based on the aforementioned considerations, the main modifications caused by thermal aging on fique fabric-reinforced epoxy matrix composites were investigated. Figure 1 presents schematically the structure of a fique-fiber woven fabric-reinforced epoxy composite, proposed in the present work, before aging and after three thermal agings at different times of exposure, making clear the changes in structure and rupture mechanisms. In view of this, the polymer ability to resist thermomechanical degradation, defined as its thermal stability, was studied [10]. This property is evaluated by the temperature at which polymer degradation begins to be perceptible, by the formed products and by the reaction kinetics [11]. During the degradation process, color changes are observed followed by the formation of solid and gaseous products [6]. Generally, these reactions can be divided into two main groups: (i) reactions occurring without main chain breakage; and (ii) reactions with main chain rupture, which is normal thermal degradation.

Indeed, some general degradation processes may involve a combination of two or more individual mechanisms [13,14,15,16,17,18,19,20,21,22,23,24,25,26,27]. Some references that considered high temperature aging and its effects are shown below in Table 1.

In the absence of light and at room temperature (RT), most polymers are stable for long periods of time [2]. However, as reported by Qin et al. [28], under sunlight the oxidation rate of polymers is accelerated and this effect can be exacerbated by the presence of atmospheric pollutants, as well as nitrogen and sulfur oxides, which are frequent components of the industrial atmosphere. In this case, the degradation occurs due to the phenomenon of photolysis, which is a chemical decomposition caused by light and photo-oxidation [2,20,27]. Natural fibers can undergo degradation due to biological agents, acid and alkaline media, moisture absorption, radiation and temperature. According to Balla et al. [12], the lignocellulosic components of natural fibers respond in different ways to the aforementioned media. Thus, considering the above, it is important to carry out aging studies under separate conditions, making it possible to understand the phenomena that are produced by each mechanism involved in the aging.

Thermal analysis is often used to describe the analytical technique that investigates the behavior of a sample as a function of temperature variation [29]. This makes it possible to qualitatively and quantitatively characterize a large number of thermal occurrences over a wide temperature range. As stated by Fan et al. [6] and Azwa and Yousif [8], the exposure to elevated temperature induced thermal stresses between fibers and matrices due to differential thermal expansion between polymer matrices and fibers. This differential thermal expansion of the fiber and matrix may lead to the formation of microcracks at the fiber/matrix interface. In a previously reported work [10], the aging mechanisms were therefore dominated by the effects of the unreacted resin components present in the laminate. Hemicellulose loss and oxidation were pointed out as the main mechanisms for the microstructure changes that occur during thermal aging of fique-fiber woven fabric-reinforced epoxy composites. All samples of fique-fiber woven fabric, plain epoxy and fique-fiber woven fabric composites revealed physical aspect changes at the macrostructural level, exhibiting a formation of a brownish layer at the sample surface [10].

Additionally, ballistic vests, manufactured based on high-performance polymers, have an expiration date fixed from the date of manufacture, and must be permanently disposed of or destroyed after this period, causing financial loss to the country [30,31]. The materials presented in this article have already been studied with a proposal for partial replacement of the common ballistic panel [32,33,34,35]. These components of the ballistic panels determine the level of vest protection and are planned according to the ballistic threat they must withstand [31]. Thus, in the present work, the aging of these proposed ballistic panels was considered.

The present work details the results from an extensive work program that was undertaken to develop thermal accelerated aging tests for ballistic composites that could be used to partially replace some of the expensive and unsustainable synthetic fibers.

## 2. Materials and Methods

### 2.1. Materials and Composite Processing

Fique-fiber woven fabric, illustrated in Figure 2, was supplied by co-author Henry Colorado and considered as composite’s reinforcement. These fique-fiber woven fabrics were cut into dimensions of 15 × 12 cm and placed in an oven at 70 °C for 24 h to remove moisture absorbed by the fiber.

Epoxy polymer composed of diglycidyl ether of bisphenol A (DGEBA) and the hardener triethylenetetramine (TETA) was used as matrix. Both DGEBA and TETA were supplied by Epoxyfiber, Rio deJaneiro, Brazil. The mixture of these two components was made with a stoichiometric ratio phr 13, recommended by manufacturer. Composite plates with 15, 30, 40 and 50 vol% of fique fabric, Table 2, were produced in a steel mold with an internal volume of 180 cm³ (150 × 120 × 10 mm³) by hand lay-up process. For epoxy, a density of 1.1 g/cm³ was adopted [36] and for the fique-fiber woven fabric 0.67 g/cm³ [32].

To facilitate the plate demolding, a layer of silicone grease was applied to metallic mold followed by alternating layers of epoxy and fique-fiber woven fabric. After complete accommodation of components, the metallic mold was closed applying a load of 5 tons (3 MPa) for 24 h. The percentage of reinforcement used in the thermal aging study was based on previous works [33,34,35], where the integrity maintained in the composite of 40% in volume of fique-fiber woven fabric reinforcement was observed, guaranteeing superior ballistic protection in comparison with the use of Kevlar^®^, taking into account the energy absorption capacity. Likewise, the multilayered ballistic system (MBS) with the composite reinforced with 40% fabric, although not the one that presented the lowest trauma depth, exhibited better ballistic performance taking into account the cohesion of material and reliability level.

### 2.2. Accelerated Thermal Aging

The fique-fiber woven fabric-reinforced epoxy composite and plain epoxy were subjected to accelerated aging test performed in air chambers (Nova Instruments, Brazil) at 170 °C for progressively increasing time lengths (0, 3, 5 and 10 days). The aging temperature of 170 °C was selected as the limit temperature before degradation in both the epoxy [9,37] and the fique-fiber woven fabric. The continuous use of temperatures in an unstressed state generally vary from 70 °C up to 200 °C, according to Biron [29]. Table 3 lists the nomenclatures used for the samples in this work.

0 h (no aging)—reference sample;Aging temperature: 170 °C;Aging time: 72, 120 and 240 h.

**Table 3 polymers-15-03761-t003:** Environmental conditions of aging and aged materials [10].

Material	Aging Times (Hours)			
	0	72	120	240
Plain epoxy	PE-T0	PE-T72	PE-T120	PE-T240
Composite	FC-T0	FC-T72	FC-T120	FC-T240

### 2.3. Thermogravimetric Analysis (TGA)

TGA, using curves of TG and derivative DTG, was performed to obtain the thermal properties of composites and epoxy for different evaluation groups. The test conditions were performed under a nitrogen atmosphere at a flow of 50 mL/min, varying from 30 °C to 700 °C, at a heating rate of 10 °C/min. TA Instruments equipment (New Castle, DE, USA) was used for this analysis.

### 2.4. Thermomechanical Analysis

Thermomechanical analysis was carried out with Shimadzu equipment (Barueri, SP, Brazil), model TMA-60, in order to obtain the thermal expansion coefficients of the composites and epoxy for the different evaluation groups. The samples were prepared in accordance with the ASTM E831 standard [38], being placed on a quartz support, under a nitrogen atmosphere, with a temperature range of 25 °C to 200 °C and a fixed compression load of 10 gf.

### 2.5. Differential Scanning Calorimetry Analysis

The use of differential scanning calorimetry technique aimed to obtain knowledge of the glass transition and to evaluate the cure of epoxy resin temperature, as well as the enthalpies involved in the degradation process of the material caused by aging. The analysis was carried out in a Shimadzu DTG-60H model (Barueri, SP, Brazil). Samples were prepared in accordance with ASTM D3418 standard [39], with masses of ≈5 mg, and were subjected to a single heating cycle from 25 °C to 400 °C, at a rate of 10 °C/min, under a nitrogen flow of 50 mL/min.

### 2.6. Colorimetry Analysis: CIE Lab

Colorimetric analysis was performed to evaluate surface color changes using a chromometer, according to the CIE *L* a* b** color system. The colorimetry analysis aimed at evaluation using the coordinates of this method. The *L** represents the luminosity value, where the darkest black is *L** = 0 and the brightest white is *L** = 100. The green–red and blue–yellow components are represented by the chromaticity coordinates *a** and *b**, respectively. The red and yellow components are shown in the positive direction, while the green and blue components are shown in the negative direction. Total color changes (Δ*E*) were calculated as described in ISO 7724 [40], according to Equation (Equation 1), where Δ*L*, Δ*a* and Δ*b* are the differences between the initial (non-weathered sample) and final (weathered sample) values of *L**, *a** and *b**.
(1)$$\Delta E=\sqrt{{\left(\Delta L\right)}^{2}+{\left(\Delta a\right)}^{2}+{\left(\Delta b\right)}^{2}}$$

The whiteness index, Equation (Equation 2), is a numerical indicator used to indicate the degree of whiteness. In the CIE Lab color space, two of the axes are perceptually orthogonal to the luminosity. Hue can be calculated along with chroma, Equation (Equation 3), transforming coordinates *a* and *b* from rectangular to polar form. Hue is the angular component of the polar representation, while chroma is the radial component. The colorimetric pattern tests of the plates were carried out in a portable colorimeter model WR-10QC, ARTBULL, CN (Barueri, SP, Brazil).
(2)$$W_{i}=100-\sqrt{{\left(100-L\right)}^{2}+a^{2}+b^{2}}$$
(3)$$C_{ab}^{*}=\sqrt{a^{2}+b^{2}}$$

## 3. Results and Discussion

### 3.1. Thermogravimetric Analysis

Thermogravimetric analysis (TGA) coupled with differential thermal analysis (DTA) and derived thermogravimetric (DTG) analysis was carried out to determine the materials’ thermal stability, and to find evidence of any exothermic or endothermic process in fique-fiber woven fabric-reinforced epoxy composites with different percentages (Figure 3) and aged materials (Figure 4 and Figure 5). As a preliminary study, analyses were carried out on epoxy matrix composites reinforced with 15%, 30%, 40% and 50% fique-fiber woven fabric reinforcement. Figure 3a and Table 4 present a small weight loss (<4%) occurring at a temperature around 100 °C. The greatest weight loss, about 59%, occurred between 260 °C and 400 °C for the FC40/E60 composite. It was also noticed that the weight loss rate during degradation at up to 100 °C of FC30/E70 and FC50/E50 was lower than FC15/E85, once their thermal stabilities were increased and the degradation process shifted to a higher temperature, which can be evidenced by the transposition of the DTA peak to higher temperatures.

The DTA curve in Figure 3b only shows a symmetrical and uniform peak related to a maximum rate of weight loss at 322 °C, 313 °C, 333 °C and 332 °C for FC15/E85, FC30/E70, FC40/E60 and FC50/E50, respectively. In general, thermal decomposition causes endothermic peaks and oxidative decomposition causes exothermic peaks, as can be seen for the composites. As shown in Figure 3c, in the region of the TGA curve drop, where weight loss is most apparent, the DTG curve shows an expressive peak at between 311 °C and 346 °C. Fique-fiber woven fabric-reinforced epoxy composites have been shown to have weight loss in up to two stages. It should be noted that the residues at 500 °C were 30, 39, 29 and 33%, respectively.

With the purpose of understanding whether or not the thermal stability of the studied composites has been obtained, it is necessary to observe the thermal behavior of the polymeric resin and the fiber in its singularities. With this in mind, Bard et al. [41] report the onset temperature of the large weight loss for the DGEBA/TETA system at around 250 °C [42]. Pistor et al. [43] mention that the first stage, at between 150 and 320 °C, is related to the presence of low molecular weight fractions that do not participate in the curing process. And the second stage of weight loss (<320 °C) is related to the presence of allylic ethers formed by the amine or ester bonds from the dehydration of the secondary alcohol present in the structure of the epoxy. For the fique-fiber contained as reinforcement, Sánchez et al. [44] reports that at a temperature of 280 °C there is a marked reduction in weight and this event may be related to the decomposition of the fiber structure. This occurs due to the rupture of the macromolecular chains and decomposition process of hemicellulose and cellulose. Unlike hemicellulose, cellulose is a linear polymer without branches and a higher order, which gives it a higher thermal stability. It is emphasized that the maximum working temperature for the studied composites considered was 200 °C, once the moment represented by the beginning of the sudden weight drop in the TGA curve is fixed as the thermal stability temperature [45].

Taking into account Figure 4a, for the aged and non-aged DGEBA/TETA system, it can be noticed that weight loss for plain epoxy is almost negligible, around 2%, up to approximately 200 °C. TGA onset temperatures are almost the same for all samples, in the range of 278–301 °C. One can observe that PE-T240 showed the lowest weight loss during the onset of sudden weight loss and PE-T72 showed the highest onset temperature, as can be seen in Table 5. In comparison to PE-T0, it was observed that the T_g_^TGA^ of PE-T72 and PE-T120 was higher. During the TGA curve drop, four peaks can be seen in DTA curves for PE-T0, PE-T120 and PE-T240, and three peaks for PE-T72 (Figure 4b). The first, considered as a shoulder, is related to the evaporation of moisture and low molecular weight oligomers, between 79 and 126 °C. The second peak is related to the main decomposition that occurs in the DGEBA/TETA system, i.e., associated with rupture and degradation of polymeric chains, between 297 and 340 °C. The third (431–435 °C) and fourth (561–571 °C) peaks are secondary breakdowns. The fourth peak, in particular, may be indicative of the complexity of composition formed in PE-T0, either because it is a post-cure, or due to prior degradation of the components for PE-T120 and PE-T240. Because the degradation phenomena occur in a heterogeneous manner and form simultaneously, the chain scission gives rise to several products, such as combustible gases, allylic alcohol, acetone and various hydrocarbons [43].

Regarding fique-fiber woven fabric-reinforced epoxy composite, slight weight loss can be seen, around 3%, up to approximately 200 °C. Figure 5a shows that TGA onset temperatures are slightly decreased as exposure to high temperatures increases, between 278 and 296 °C. The greatest weight loss of the composite is in FC-T72, with a residue of only 2%, followed by FC-120, with a residue of approximately 4%, up to 700 °C. During the TGA curve drop, three peaks can be seen in DTA curves under all conditions, as one can see in Figure 5b. The first peak, between 93 and 113 °C, occurs lightly and is associated with the removal of moisture when the sample was heated and evaporation of moisture and low molecular mass oligomers, as also seen in the DGEBA/TETA system. The second peak, between 316 and 332 °C, is associated with the decomposition of hemicelluloses [46]. The third and largest peak occurs between 565 and 633 °C, and is related to the degradation of resin and its solvents [47], as well as the cellulose and lignin degradation. These statements are based on the non-stability of the curves at temperatures above 400 °C and the amplitude of the peak presented in the aforementioned range. Lignin was the most difficult to decompose; its decomposition occurred slowly at temperatures higher than 400 °C [12,46]. The enthalpy variation decrease compared to the DGEBA/TETA system samples can be observed clearly (Figure 4b).

It is important to note that aged composites had fewer intense initial peaks, which proves the effects of the fique-fiber woven fabric reinforcement. This is because the thermal properties of the lignocellulosic fibers, as fique fiber, are mainly influenced by their composition, i.e., cellulose, hemicelluloses and lignin content [12]. In this case study it was clear that hemicelluloses are associated with moisture content [46]. It is noteworthy here that the literature reports the following contents of lignin, hemicellulose and cellulose for fique fiber, respectively: 6.81–16.6% [48], 22.1–27.1% [48,49], and 18.27–70% [48,49].

### 3.2. Thermomechanical Analysis

Figure 6 presents the TMA curves obtained for fique-fiber woven fabric-reinforced epoxy matrix composites. The glass transition temperatures were obtained by TMA (T_g_^TMA^) in the order of approximately 54 °C for FC15/E85, 62 °C for FC30/E70, 60 °C for FC40/E60 and 57 °C for FC50/E50. As the temperature increased, the material expanded slightly. All the thermal expansion curves were dispersed; however, the total variations were small, especially for FC40/E60, i.e., very close to the detection limit of the equipment; therefore, the reading accuracy decreased. The coefficient of linear thermal expansion (CLTE) obtained through three different temperature ranges is shown in Table 6.

Figure 7 shows TMA curves for aged and non-aged DGEBA/TETA system and fique-fiber woven fabric-reinforced epoxy composites. The CLTE of the DGEBA/TETA system and fique-fiber woven fabric-reinforced epoxy composites were calculated by measuring their deformation within a specific or useful temperature range. Table 7 presents the data contained in Figure 7, in which all materials showed thermal expansion.

Increases in CLTEs were noted before and after T_g_^TMA^, both for PE-T0 and FC-T0, as well as after three periods of thermal aging. The T_g_^TMA^ also increased after exposure to high temperatures. These increases may be related to the material stiffening. A peak observed at around 96–144 °C in the DGEBA/TETA system indicates the T_g_^TMA^ of the epoxy matrix. As listed in Table 7, the CLTE above T_g_^TMA^ of PE-T0 was 182 × 10${}^{-6}$ °C and this value decreased to 148 × 10${}^{-6}$ °C for PE-T120. This value was even smaller than that in the case of PE-240 (i.e., 187 × 10${}^{-6}$ °C).

In addition, composites are more sensitive to temperature variations as a result of thermal stresses induced between the fibers and polymeric matrix. This occurs due to the different CLTEs between the matrix and reinforcement material. At high temperatures, differential thermal expansion of the fiber and polymer matrix can lead to the formation of microcracks at the fiber/matrix interface, making it also susceptible to aggressive reactions, and leading to degradation of both fiber and matrix. This affects the integrity of composites, as it is through the interface that thermal and mechanical loads are transferred from the matrix to the fibers [50]. According to Chee et al. [51] each composite shows contraction and expansion in its dimension when subjected to temperature changes and is strongly influenced by the orientation and direction of the fibers.

### 3.3. Differential Scanning Calorimetry Analysis

DSC curves of fique-fiber woven fabric-reinforced epoxy composites were presented in previously work [52]. It was suggested that the variation in the amount of natural fibers used for reinforcement did not affect the thermal stability of the composite, once any variation in enthalpy was observed. The DSC curves for aged and non-aged DGEBA/TETA system and fique-fiber woven fabric-reinforced epoxy composites are shown in Figure 8. In general, first-order transitions are observed as well defined peaks, while second-order transitions are variations in the heat flow curve [53]. The glass transition temperature determined by DSC (T_g_^DSC^) is observed as a slight change in the slope of the curve, and is shown in Figure 8 and described in Table 8.

It can be seen that the T_g_^DSC^ related to the DGEBA/TETA system was reduced after the first exposure to thermal aging, with an increase in T_g_^DSC^ also being observed as exposure increased. The T_g_^DSC^ of the composite had a reduction after the first exposure to thermal aging and the same trend was observed for the DGEBA/TETA system. Jesuarockiam et al. [54] observed the initial endothermic peak at around 60–70 °C and attributed it to epoxy T_g_^DSC^. PE-T240 showed a higher T_g_^DSC^ value compared with the others. This was due to the presence of high NH groups that had accelerated the curing rate and extended more links to form higher dense polymer networks, and that resulted in high T${}_{g}$ [55].

The endothermic peak, which was more prominent for PE-T240, is related to moisture loss. It is possible to note that PE-T0 has several stress reliefs as temperature increases; these effects are reduced with exposure to high temperatures. As the temperature increases, the material gains enough energy and the microstructure is reorganized, displaying an exothermic peak at approximately 280 °C. It is important to point out that the double peaks, Figure 8a, at 292 °C and 316 °C, are related to residual/additional curing of the epoxy [10] and amine reactions [29]. At a higher temperature, the system has gained so much energy that the separation between the molecules is large enough to break the intermolecular interactions required for keeping the molecules together. The system lowers its viscosity and softening at around 370 °C [45]. An exothermic peak ranging from 284 to 349 °C was also observed and this peak is attributed to the effect of the curing of the DGEBA/TETA system, specifically related to decomposition of the aromatic epoxy and aliphatic amine hardener [54,56]. In the PE-T240 variant, displacement of the mentioned peak to a temperature of 349 °C is quite evident. And even in PE-T240 it is possible to notice a high reduction in energy flow compared to other conditions. The addition of further energy at higher temperatures, applied in PE-T240 before the analysis, produces the oxidative decomposition and degradation processes observed at 380 °C. The appearance of an exothermic peak at 389 °C, attributed to epoxy oxidation, and the final ascending of the DSC curve indicate sample degradation.

Figure 8 and Table 8 also present DSC curves obtained for the fique-fiber woven fabric-reinforced epoxy composite thermally aged for 72 h, 120 h and 240 h, and the control group. The endothermic peak, between 58 and 70 °C, can be attributed to moisture loss in the composite samples. It should be noted that this endothermic event was more prominent for samples exposed for the longest time to high temperatures, indicating a possible epoxy matrix weakening, facilitating entry of moisture, or even water, into the material structure. The first exothermic peak, at around 284 °C, was reduced with exposure to high temperatures. An exothermic peak was noted, approximately at 350 °C, indicating the beginning of hemicellulose degradation. It was expected that cellulose degradation would start to occur from 400 °C, also characterized by an endothermic event, but this was not observed. The second exothermic high, with a double peak, or even peak broadening, was observed in all composite samples, which may be related to fique-fiber degradation and DGEBA/TETA system oxidation. It stands out that the energy flow was greatly increased in the FC-T72 variant.

### 3.4. Colorimetry Analysis

Figure 9a shows that the DGEBA/TETA system and fique-fiber woven fabric-reinforced epoxy composite became darker with increasing aging and treatment time. Taking the controls groups, PE-T0 and FC-T0, as a reference, the following colorimetric alterations can be seen. Figure 9b shows a reduction of approximately 44% in luminosity (L*) for PE-T72, 43% for PE-T120 and 41% for PE-T240. Regarding composites, the luminosity parameters have a greater reduction of 51% for FC-T72, 49% for FC-T120 and 47% for FC-T240. The parameter that varied the most was a*, especially for the DGEBA/TETA system, showing an increase of around 22 times for PE-T72, 17 times for PE-T120 and 6 times for PE-T240, indicating the samples reddening. The changes in parameter a* for the composite were equivalent to 61% for FC-T72, 14% for FC-T120 and 12% for FC-T240. The b* parameter, which indicates the samples yellowing in a positive direction, also showed an increase of 11 times for PE-T72, 10 times for PE-120 and 2 times for PE-T240.

The color changed exhibited the formation of a brownish layer at the sample surface [10]. As one can observe in Figure 9c, the E${}_{ab}$ parameter increased after thermal aging for both DGEBA/TETA system and fique-fiber woven fabric-reinforced epoxy composites. This increase ranged between 41 and 46% compared to PE-T0, and between 43 and 48% compared to FC-T0. The whiteness index also showed an increase, in the range of 41–43% for the DGEBA/TETA system and 55–59% for the fique-fiber woven fabric-reinforced epoxy composites, both after thermal aging. The behavior of the chroma parameter (C${}_{ab}$) for the DGEBA/TETA system is highlighted, showing high variation. Increases were observed of 21 times for PE-T72, 16 times for PE-T120 and 5 times for PE-T240. In the study carried out by Masek and Latos-Brozio [57] with polyhydroxyalcaonate, chroma and brightness parameters were the most affected by aging. Significant color changes were observed by the authors after thermal aging combined with oxidation and they attributed this fact to the reaction of OH groups during oxidation.

## 4. Summary and Conclusions

In the present work, the thermal and colorimetric behavior of the non-aged and aged DGEBA/TETA system and epoxy composites reinforced with fiber fabric were, for the first time, investigated. The main conclusions are:TGA carried out on non-aged epoxy matrix composites reinforced with 15%, 30%, 40% and 50% fique-fiber woven fabric reinforcement showed a small weight loss (<4%) that occurred at a temperature around 100 °C. The greatest weight loss, around 59%, occurred between 260 °C and 400 °C for the FC40/E60 composite. The DTG curve showed a significant peak between 311 °C and 346 °C. Aged composites presented less intense initial peaks for the DGEBA/TETA system (plain epoxy), which proves the reinforcement effects of fique-fiber woven fabric. This is because the thermal properties of lignocellulosic fibers are mainly influenced by their composition, i.e., cellulose, hemicelluloses and lignin content.TMA performed for the non-aged composite of epoxy matrix reinforced with fique-fiber woven fabric showed a T_g_^TMA^ between 54 and 62 °C. As the temperature increased, the material expanded slightly. All thermal expansion curves were dispersed but the total variations were small, mainly for FC40/E60. Additionally, aged composites are more sensitive to temperature variations as a result of thermal stresses induced between the fibers and the polymeric matrix. At high temperatures, the differential thermal expansion of the fiber and the matrix could lead to the formation of microcracks at the fiber/matrix interface, making it also susceptible to aggressive reactions, and to the degradation of both fiber and matrix.DSC analysis curves of thermally aged and non-aged DGEBA/TETA system and fique-fiber woven fabric-reinforced epoxy composites suggest that the variation in the amount of natural fibers used for reinforcement did not affect the thermal stability of the composite once any variation in enthalpy was observed. In PE-T240 it is possible to notice a high reduction in energy flow compared to other conditions. The addition of further energy at higher temperatures, applied in PE-T240 before the analysis, produces oxidative decomposition and degradation processes observed at 380 °C.Colorimetry analysis showed a reduction of approximately 40% in luminosity (L*). Regarding composites, the luminosity parameters have a greater reduction of 50%. Additionally, highlighting the chroma parameter for the DGEBA/TETA system, it showed high variation, 21 times for PE-T72, 16 times for PE-T120 and 5 times for PE-T240.

In summary, significant changes in the color of the DGEBA/TETA system and the fique-fiber woven fabric-reinforced epoxy composites were observed after thermal aging combined with a suggested oxidation. It was also observed that the thermal behavior of the DGEBA/TETA system showed greater resistance after thermal exposure. The composites, on the other hand, were more sensitive to temperature variations as a result of thermal stresses induced between the fique fibers and the epoxy matrix.

## Figures and Tables

**Figure 1 polymers-15-03761-f001:**
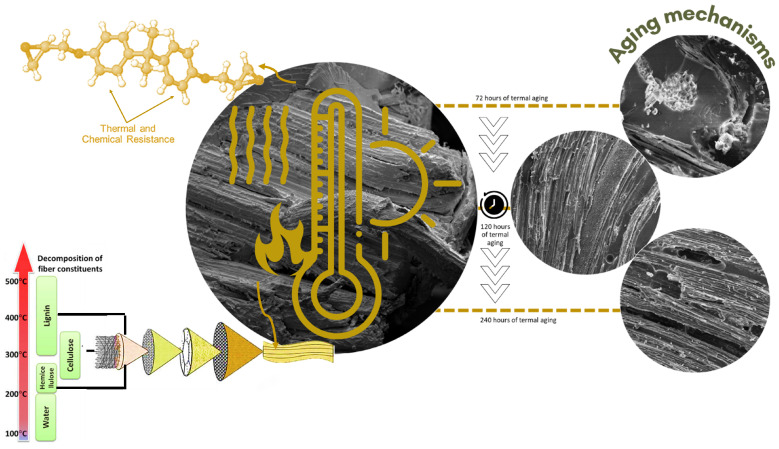
Schematic showing the degradation temperature ranges for different constituents of natural fibers, and the chemical structure of bisphenol A diglycidyl ether epoxy resin (DGEBA) and fique-fiber woven fabric-reinforced composite after aging (adapted from Refs. [10,12]).

**Figure 2 polymers-15-03761-f002:**
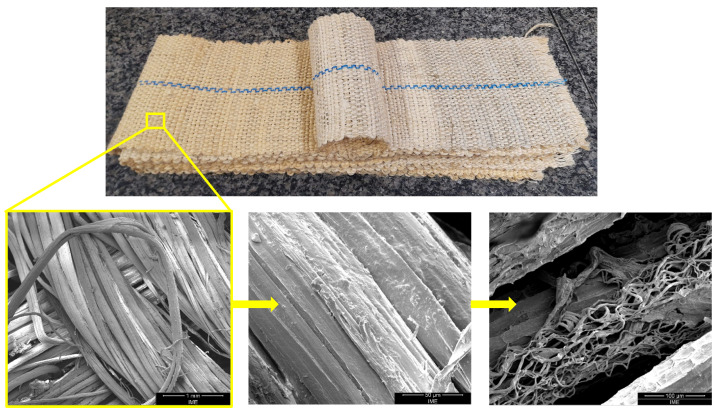
Architecture of fique-fiber woven fabric.

**Figure 3 polymers-15-03761-f003:**
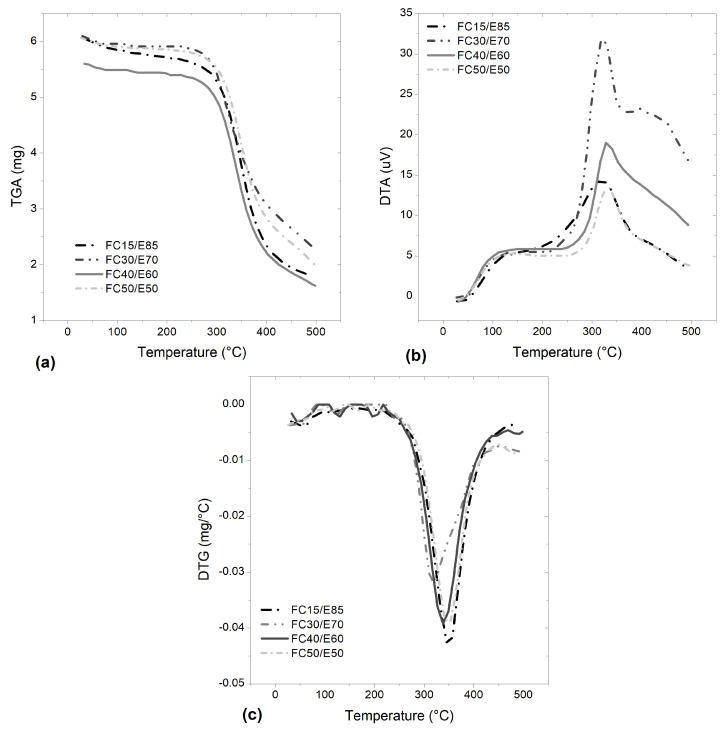
(**a**) Thermogravimetric analysis (TGA) coupled with (**b**) differential thermal analysis (DTA) and (**c**) derivative thermogravimetry (DTG) carried out on non-aged epoxy matrix composites with 15%, 30%, 40% and 50% fabric reinforcement.

**Figure 4 polymers-15-03761-f004:**
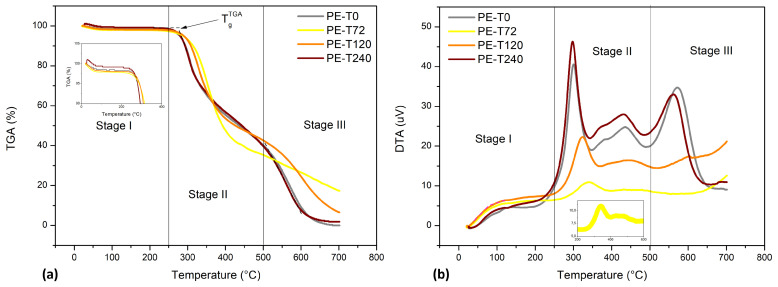
(**a**) Thermogravimetric analysis (TGA) coupled with (**b**) differential thermal analysis (DTA) carried out on aged and non-aged DGEBA/TETA system.

**Figure 5 polymers-15-03761-f005:**
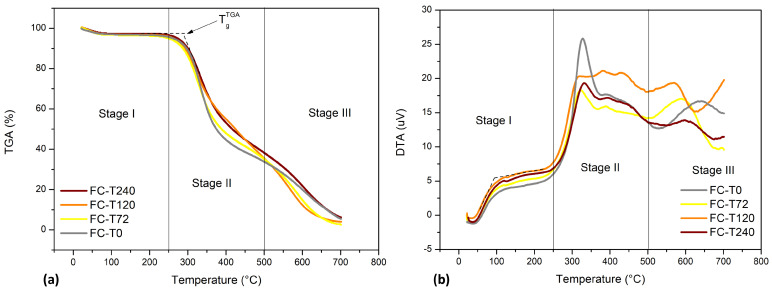
(**a**) Thermogravimetric analysis (TGA) coupled with (**b**) differential thermal analysis (DTA) carried out on thermally aged and non-aged fique-fiber woven fabric-reinforced epoxy composites.

**Figure 6 polymers-15-03761-f006:**
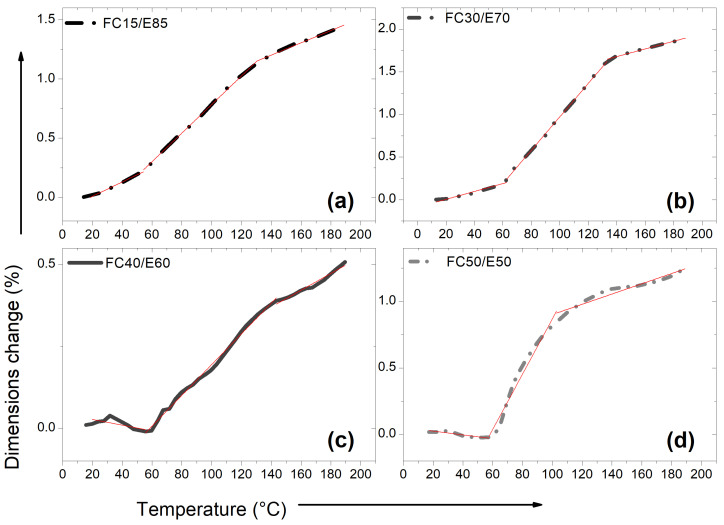
Dimensional change versus temperature change for (**a**) FC15/E85, (**b**) FC30/E70, (**c**) FC40/E60 and (**d**) FC50/E50.

**Figure 7 polymers-15-03761-f007:**
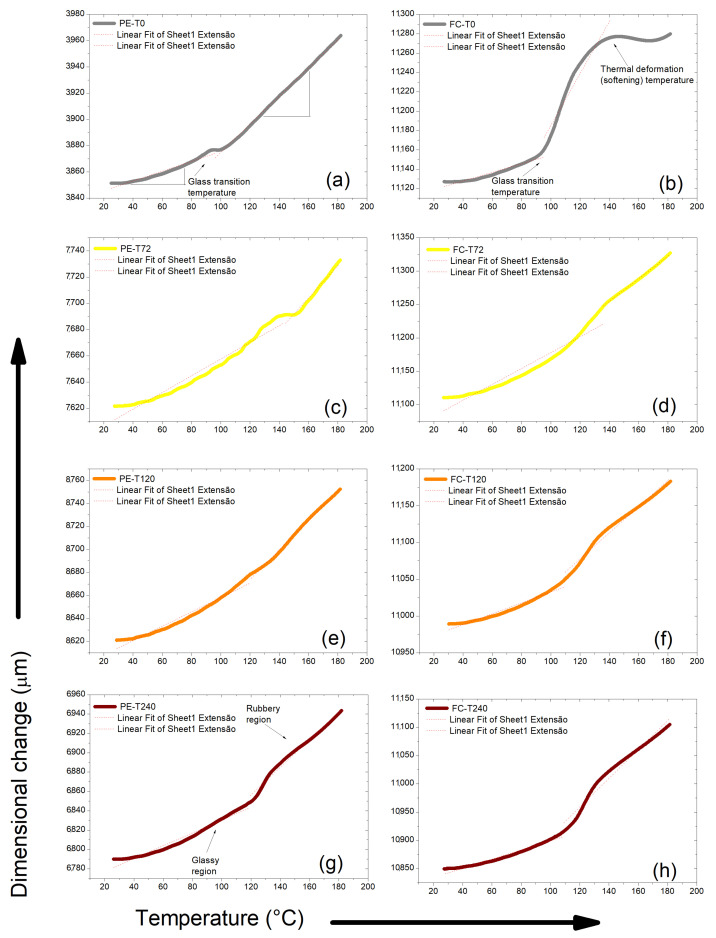
Thermomechanical analysis (TMA) carried out on thermally aged and non-aged DGEBA/TETA system and fique-fiber woven fabric-reinforced epoxy composites: PE−T0 (**a**), FC−T0 (**b**), PE−T72 (**c**), FC−T72 (**d**), PE−T120 (**e**), FC−T120 (**f**), PE−T240 (**g**) and FC−T240 (**h**).

**Figure 8 polymers-15-03761-f008:**
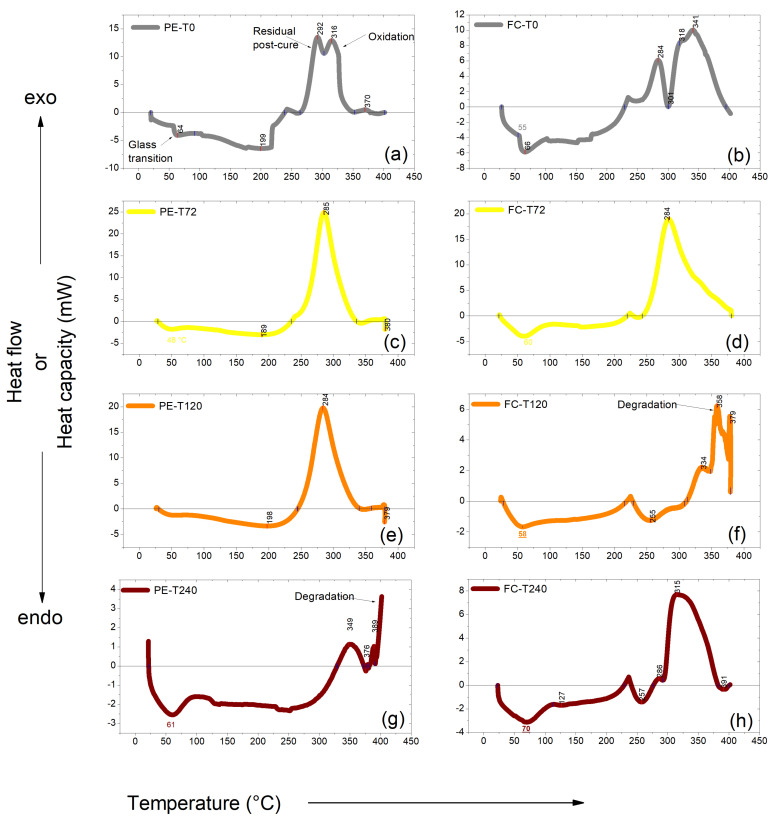
DSC curves of thermally aged and non-aged DGEBA/TETA system and fique-fiber woven fabric-reinforced epoxy composites: PE−T0 (**a**), FC−T0 (**b**), PE−T72 (**c**), FC−T72 (**d**), PE−T120 (**e**), FC−T120 (**f**), PE−T240 (**g**) and FC−T240 (**h**).

**Figure 9 polymers-15-03761-f009:**
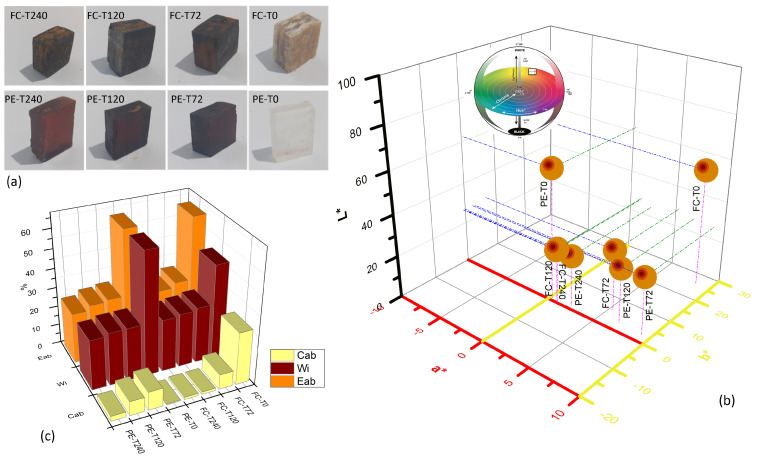
CIE L* a* b* colorimetric parameter values measured for the investigated aged and non-aged DGEBA/TETA system and fique-fiber woven fabric-reinforced epoxy composites: (**a**) specimen color changes, (**b**) changes in color coordinates of CIELAB, (**c**) main parameters during thermal aging.

**Table 1 polymers-15-03761-t001:** Summary of some references that considered high temperature aging and its effects.

Type of Aging	Fiber	Matrix	Effects	Reference
High temperature, pH, natural	Glass	Vinyl ester	Significant degradation, up to 70% at high temperature (30 days), weakening of the fiber/matrix interface.	Hota et al. [2]
High temperature, moisture	Flax	Epoxy	Degradation of mechanical properties is attributed to fiber/matrix detachment and cohesive failure in fiber bundles.	Koolen et al. [3]
High temperature, moisture	Sisal	Mortar	Cyclical moisture changes at relatively high temperatures accelerate natural fiber degradation in the cement matrix more effectively than aggressive static conditions.	Wei and Meyer [4]
High temperature, moisture	-	Epoxy	Autoxidation of the amine molecular crosslinker—oxidation of the amine groups starts in the same way: via the formation of radical amino cations by the one-electron oxidation of the incorporated amine groups.	Morsch et al. [11]
High temperature, humidity, water immersion	Carbon	Epoxy	Prolonged dry thermal aging at 170 °C causes a reduction in bending properties.	Birger et al. [5]
High temperature	Carbon	Epoxy	Progressive deterioration of the matrix and fiber/matrix interfaces, in the form of chain scissions, oxidation of carbon elements, loss of mass, cracks, significant decrease in thermal conductivity.	Fan et al. [6]
High temperature	Carbon	Epoxy	The degree of cross-linking of the resin was increased, mass loss attributed to absorbed moisture, increased glass transition temperatures of epoxy resin systems.	Souza et al. [7]
High temperature	Kenaf	Epoxy	Increasing exposure time causes greater weight loss of composites only up to 150 °C. Fiber/matrix detachment was observed in degraded samples due to fine cracks from 150 °C onwards, implying mechanical degradation of the composites. Physical shrinkage at 250 °C.	Azwa and Yousif [8]
High temperature	-	Epoxy	Samples aged at 170 °C, 150 °C and 130 °C show a glass transition at about 3, 15 and 60 days, respectively.	min Pei et al. [9]

**Table 2 polymers-15-03761-t002:** Different percentages of reinforcement of fique-fiber woven fabric in epoxy composites evaluated.

Material	Reinforcement Content	Epoxy Content
FC15/E85	15 vol%	85 vol%
FC30/E70	30 vol%	70 vol%
FC40/E60	40 vol%	60 vol%
FC50/E50	50 vol%	50 vol%

**Table 4 polymers-15-03761-t004:** T_g_^TGA^, weight loss and DTA parameters for different percentages of reinforcement of fique-fiber woven fabric in epoxy composites.

Material	T_g_^TGA^ (°C)	Weight Loss (%)		Peak	Residue at	Heat
		at 100 °C	up 100 °C	(°C)	500 °C (%)	
FC15/E85	281	4.06 (0.25 mg)	56.58 (3.46 mg)	322	1.8	2.37 J (386 J/g)
FC30/E70	301	2.09 (0.13 mg)	45.55 (2.76 mg)	313	2.3	2.95 J (485 J/g)
FC40/E60	291	2.95 (0.16 mg)	58.84 (3.32 mg)	333	1.6	1.36 J (239 J/g)
FC52/E50	299	2.69 (0.16 mg)	48.52 (2.93 mg)	332	1.9	1.90 J (314 J/g)

**Table 5 polymers-15-03761-t005:** T_g_^TGA^, weight loss and DTA parameters for different percentages of reinforcement of aged and non-aged DGEBA/TETA system and fique-fiber woven fabric epoxy composites.

Material	T_g_^TGA^	Weight Loss (%)/ DTA Peak (°C)			Residue	Residue
		Stage I	Stage II	Stage III	at 500 °C (%)	at 700 °C (%)
PE-T0	282	6/126	62/300; 435	96/571	40	0
PE-T72	301	8/83	58/340; 436	72/-	35	17
PE-T120	294	6/79	47/323; 433	77/602	42	6
PE-T240	278	5/88	60/297; 431	96/561	40	2
FC-T0	296	10/113	54/327	89/633	33	5
FC-T72	286	10/108	51/323	95/587	34	2
FC-T120	280	8/93	48/316	93/565	35	4
FC-T240	278	6/104	49/332	89/597	38	6

**Table 6 polymers-15-03761-t006:** The thermal expansion coefficients for fique-fiber woven fabric-reinforced epoxy composites.

Material	T_g_^TMA^ (°C)	CLTE_1_ (×10${}^{-3}$ °C${}^{-1}$) (R²)	CLTE_2_ (×10${}^{-3}$ °C${}^{-1}$) (R²)	CLTE_3_ (×10${}^{-3}$ °C${}^{-1}$) (R²)
FC15/E85	54	6.06 (0.99)	12.0 (0.99)	5.19 (0.98)
FC30/E70	62	4.64 (0.94)	19.6 (0.99)	4.49 (0.99)
FC40/E60	60	−0.86 (0.41)	4.69 (0.99)	2.56 (0.96)
FC50/E50	57	−1.42 (0.80)	20.7 (0.96)	3.85 (0.92)

**Table 7 polymers-15-03761-t007:** T_g_^TMA^ and CLTE of thermally aged and non-aged DGEBA/TETA system and fique-fiber woven fabric-reinforced epoxy composites.

Material	T_g_^TMA^	CLTE (×10${}^{-5}$ °C${}^{-1}$) before T${}_{g}$	R²	CLTE (×10${}^{-4}$ °C${}^{-1}$) after T${}_{g}$	R²
PE-T0	96	7.637 ± 0.006	0.96	1.8253 ± 0.0003	0.99
PE-T72	114	8.336 ± 0.006	0.95	1.8161 ± 0.0008	0.99
PE-T120	120	7.262 ± 0.006	0.96	1.4877 ± 0.0004	0.99
PE-T240	119	10.443 ± 0.002	0.97	1.8741 ± 0.0002	0.99
FC-T0	94	3.890 ± 0.005	0.93	2.3071 ± 0.0035	0.93
FC-T72	136	10.970 ± 0.0115	0.93	1.5101 ± 0.0003	0.99
FC-T120	109	6.734 ± 0.007	0.95	1.3497 ± 0.0003	0.99
FC-T240	109	7.637 ± 0.006	0.96	1.8253 ± 0.0003	0.99

**Table 8 polymers-15-03761-t008:** T_g_^DSC^ and chemical reactions of thermally aged and non-aged DGEBA/TETA system and fique-fiber woven fabric-reinforced epoxy composites.

Material	T_g_^DSC^	Endothermic Peak (°C)/ Heat (J/g)	Exothermic Peak (°C)/ Heat (J/g)
PE-T0	57	64/175	292; 316/618
PE-T72	48	48/60	285/906
PE-T120	54	54/40	282/720
PE-T240	61	61/142	349/224
FC-T0	55	66/231	284/144 and 318; 341/562
FC-T72	60	60/199	284/1090
FC-120	58	58/109	334; 358; 379/145
FC-240	70	70/210	315/436

## Data Availability

The data presented in this study are available on request from the corresponding author.

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
