# Peer review of "Thermal and Colorimetric Parameter Evaluation of Thermally Aged Materials: A Study of Diglycidyl Ether of Bisphenol A/Triethylenetetramine System and Fique Fabric-Reinforced Epoxy Composites"

_polymers, 2023, doi:10.3390/polym15183761_

Round 1

Reviewer 1 Report

In this manuscript, Oliveira and co-authors studied the thermal and colorimetric properties of fique-fiber woven fabric reinforced epoxy composites at different mixing ratio and aging durations. The manuscript is highly logical, well organized and properly written. The experiment is relatively simple but generally well discussed. Overall, I believe it deserves to be published in Polymer. I only have some minor questions/suggestions for the authors.

1.  It is noticed that the epoxy matrix composites reinforced with 15%, 30%, 40% and 50% of fique-fiber woven fabric reinforcement show quite random results in TGA, DTA, and DTG characterizations without obvious trend, especially the FC30/E70 group. Could the authors discuss more about this abnormal behavior?

2. Since the authors believe thermal stress is the reason for the temperature sensitivity of the composite, I suggest adding one more experiment group PE01 & FC01 (1hr aging) to double confirm it, before the material property changes.

3. Last, in Fig.2, it would be great if the authors could further provide a SEM image for the Fique-fiber woven fabric to give a straightforward impression of this natural fiber.

Author Response

Manuscript POLYMERS-2581149

Response to Reviewers

The authors would like to thank the Reviewers for the valuable comments and suggestions that contribute to improve our manuscript. Amendments were provided accordingly and all modifications were marked as Track Changes in the revised version of the manuscript. Responses to each comment, point by point, are given below.

Reviewer #1 comments:

In this manuscript, Oliveira and co-authors studied the thermal and colorimetric properties of fique-fiber woven fabric reinforced epoxy composites at different mixing ratio and aging durations. The manuscript is highly logical, well organized and properly written. The experiment is relatively simple but generally well discussed. Overall, I believe it deserves to be published in Polymer. I only have some minor questions/suggestions for the authors.

  1. It is noticed that the epoxy matrix composites reinforced with 15%, 30%, 40% and 50% of fique-fiber woven fabric reinforcement show quite random results in TGA, DTA, and DTG characterizations without obvious trend, especially the FC30/E70 group. Could the authors discuss more about this abnormal behavior?

Response

The authors are grateful for the reviewer's assessment.

In fact, this behavior of the 30/70 sample stands out, especially in the DTA graph, where the maximums are the results of exothermic processes, that is, where there was a greater amount of heat released from the sample, causing an increase in temperature. Sometimes this fraction of reinforcement is understood as the optimal fraction, such results also point to this. However, based on other results obtained by the authors, for the ballistic application, the optimal fraction was considered as the one with 40% reinforcement, and therefore the present study focused on this fraction. But, as noted by the reviewer, the good performance of the 30/70 composite could have another good application.

  1. Since the authors believe thermal stress is the reason for the temperature sensitivity of the composite, I suggest adding one more experiment group PE01 & FC01 (1hr aging) to double confirm it, before the material property changes.

Response

To confirm this, we can cite other works developed by the authors, which have already been published, refer to thermal analyzes and show the reliability of the results. Other results already obtained were not placed here because we understand that it would be out of context, in addition to being quite extensive.

Please follow the articles:

Michelle Souza Oliveira; Fernanda Santos da Luz; Andreza Menezes Lima; Foluke Salgado de Assis; Artur Camposo Pereira; Fábio de Oliveira Braga; Sérgio Neves Monteiro; André Ben-Hur da Silva Figueiredo. Effects of thermal aging and functionalized epoxy matrix with graphene nanoplates in fique fabric-reinforced composites. Tecnol. Metal. Mater. Min., vol.20, e2758, 2023.

http://dx.doi.org/10.4322/2176-1523.20222758

Michelle Souza Oliveira, Fabio da Costa Garcia Filho, Fernanda Santos da Luz, Luana Cristyne da Cruz Demosthenes, Artur Camposo Pereira, Henry Alonso Colorado, Lucio Fabio Cassiano Nascimento, Sergio Neves Monteiro . Evaluation of Dynamic Mechanical Properties of Fique Fabric/Epoxy Composites. Materials Research. 2019; 22(suppl. 1): e20190125. http://dx.doi.org/10.1590/1980-5373-MR-2019-0125

Oliveira, Michelle Souza, Fernanda Santos da Luz, Fabio da Costa Garcia Filho, Artur Camposo Pereira, Vinícius de Oliveira Aguiar, Henry Alonso Colorado Lopera, and Sergio Neves Monteiro. 2021. "Dynamic Mechanical Analysis of Thermally Aged Fique Fabric-Reinforced Epoxy Composites" Polymers 13, no. 22: 4037. https://doi.org/10.3390/polym13224037

  1. Last, in Fig.2, it would be great if the authors could further provide a SEM image for the Fique-fiber woven fabric to give a straightforward impression of this natural fiber.

Response

A SEM image was added for better understanding of the fique-fiber woven fabric.

Reviewer 2 Report

The authors have presented an interesting set of results for the DGEBA/TETA system and fique-fiber woven fabric-reinforced epoxy composite showing their use for superior ballistic protection in multilayered ballistic systems (MBS) .

  Using various analytical methods they showed how samples heating influences mechanical parameters as indicated the thermal ranges of the structural changes.

The results are interesting however it could be also interesting to beter prove the structural changes by performing some other techiques like spectroscopic techicues or microscopic ones like  SEM techique to visualise the fabric structural evolution with temeperature.

At least it would be interesting to propose some references documenting some analysis by other authors accodring to structural changes in this type of materials.

Author Response

Manuscript POLYMERS-2581149

Response to Reviewers

The authors would like to thank the Reviewers for the valuable comments and suggestions that contribute to improve our manuscript. Amendments were provided accordingly and all modifications were marked as Track Changes in the revised version of the manuscript. Responses to each comment, point by point, are given below.

Reviewer #2 comments:

The authors have presented an interesting set of results for the DGEBA/TETA system and fique-fiber woven fabric-reinforced epoxy composite showing their use for superior ballistic protection in multilayered ballistic systems (MBS) . Using various analytical methods they showed how samples heating influences mechanical parameters as indicated the thermal ranges of the structural changes.

The results are interesting however it could be also interesting to beter prove the structural changes by performing some other techiques like spectroscopic techicues or microscopic ones like  SEM techique to visualise the fabric structural evolution with temeperature.

Response

The authors are grateful for the reviewer's assessment. Regarding the additions suggested by the reviewer, the authors are not opposed to presenting them, but there is some concern with the size of the article, as each result of the analyzes mentioned generates a complete article, however heavy and massive for reading. Therefore, if possible, we chose to present the results mentioned in the near future. There is, in figure 1 of this article, the SEM analysis performed both in the reference sample 40/60 and in the aged ones.

At least it would be interesting to propose some references documenting some analysis by other authors accodring to structural changes in this type of materials.

 Response

As requested through the good observation of the reviewer, a table was inserted with some references that document analyzes by other authors regarding structural alterations in this type of materials.
